# A Comprehensive Review on Chemical Synthesis and Chemotherapeutic Potential of 3-Heteroaryl Fluoroquinolone Hybrids

**DOI:** 10.3390/antibiotics12030625

**Published:** 2023-03-21

**Authors:** Halyna Hryhoriv, Sergiy M. Kovalenko, Marine Georgiyants, Lyudmila Sidorenko, Victoriya Georgiyants

**Affiliations:** 1Pharmaceutical Chemistry Department, National University of Pharmacy, 61002 Kharkiv, Ukraine; 2Organic Chemistry Department, Karazin National University, 61022 Kharkiv, Ukraine; 3Department of Anesthesiology Intensive Therapy and Pediatrics Anesthesiology, Kharkiv National Medical University, 61022 Kharkiv, Ukraine

**Keywords:** fluoroquinolones, synthesis, 3-heteroaryl hybrids, biological activity, antimicrobials, anticancer medicines

## Abstract

Fluoroquinolones have been studied for more than half a century. Since the 1960s, four generations of these synthetic antibiotics have been created and successfully introduced into clinical practice. However, they are still of interest for medicinal chemistry due to the wide possibilities for chemical modification, with subsequent useful changes in the pharmacokinetics and pharmacodynamics of the initial molecules. This review summarizes the chemical and pharmacological results of fluoroquinolones hybridization by introducing different heterocyclic moieties into position 3 of the core system. It analyses the synthetic procedures and approaches to the formation of heterocycles from the fluoroquinolone carboxyl group and reveals the most convenient ways for such procedures. Further, the results of biological activity investigations for the obtained hybrid pharmacophore systems are presented. The latter revealed numerous promising molecules that can be further studied to overcome the problem of resistance to antibiotics, to find novel anticancer agents and more.

## 1. Introduction

The development of synthetic antibiotics of the quinolone group started in the 1960s. Up to the 2020s, four generations of well-known and highly efficient fluoroquinolones (FQs) have been developed [1]. However, medicinal chemistry is still concerned with investigations in this area. What are the reasons for such interest and attention?

First of all, FQs possess a wide range of antibacterial activity [2,3,4]. They are prescribed for the treatment of different diseases caused by Gram-negative [5] and Gram-positive microorganisms [6], for a tuberculosis cure [7,8] in the case of rare hospital strains [5,9]. This makes it possible to use them in the therapy of bacterial pneumonia related to COVID-19, as long as the pandemic is still in progress and the special treatment has not yet been created. Moreover, new derivatives reveal antiviral [10], antifungal [11], anticancer [12,13], anti-inflammatory [14] and hypoglycemic [15] activity. That is why fluoroquinolone molecules are promising objects in the search for new biologically active compounds. In addition, the therapeutic use of FQs and other antibiotics is associated with the appearance of more dangerous and resistant strains of bacteria [16,17].

Secondly, these molecules are attractive from a chemists’ point of view due to the structure that can be modified in different ways (Figure 1). The general fluoroquinolone structure represents a bicyclic system with a substituent R_1_ at the N-1 position, a substituent R_2_ (often nitrogen-based heterocyclic moiety) at C-7, a benzene ring with or without R_3_ substituents, a carboxyl group at the 3 position that can be modified in R_4_, an oxo group at the 4 position and a fluorine atom at the 6 position [18]. Each position can be modified to change the pharmacodynamics and pharmacokinetics of the molecule, which opens broad horizons for both chemical and biological investigations [19,20,21,22].

Thirdly, we must admit that there is no way as yet to create a medicine without side effects, and fluoroquinolones are not an exception in this case. Therefore, it is necessary to improve the efficacy and safety of these medicines by synthesizing new analogues.

All these reasons together allow scientists to consider fluoroquinolones as a promising group of antibiotics, the structure of which can be modified to create novel, more active and safe medicines, with antibacterial properties and other valuable and needed types of biological activity.

Regarding the core systems that are subjected to different types of structural modifications, the first place is taken by the representatives of the second FQ generation, namely, ciprofloxacin (CF), norfloxacin (NF) and ofloxacin (OF) (Figure 2). Levofloxacin (LF) from the 3rd generation is also popular among medicinal chemists. As the second significant groups, 2nd-generation pefloxacin (PF) and 4th-generation enrofloxacin (EF) are presented. A few papers describe novel potent derivatives of 2nd-generation fleroxacin (FL), lomefloxacin (LoF), rufloxacin (RF) and 4th-generation gatifloxacin (GF) and moxifloxacin (MF).

For the 2nd-generation molecules, the most common way is to perform modifications for searching novel antimicrobials. It was backed by previous investigations and has already provided us with two more generations of FQs. All of them are effective and widely used against both Gram-negative and Gram-positive strains of bacteria, with increasing activity spectrum from generation to generation, as stated above, including multidrug-resistant bacteria. For instance, ciprofloxacin is one of the most often prescribed antibiotics in the world. Still, it is possible to improve the effectiveness safety profile as well as activity spectrum, which will be discussed further. As for other initial systems, they are often used for the expansion of the pharmacological profile, as they already have quite impressive antibacterial properties. Therefore, they are often investigated, for example, as anticancer agents.

In the current review, the results of investigations on the structural modification of FQs by introducing heterocyclic moieties into position 3 of the core system are analyzed and summarized, the approaches to the formation of heterocycles from the fluoroquinolone carboxyl group are shown, and the results of the studies of the biological activity of the obtained hybrid pharmacophore systems are presented.

The C-3 carboxylic acid group in a fluoroquinolone core can be easily modified, especially with different heterocycles via substitution and cyclization reactions that can enhance or change the basic activity or contribute to the positive effects on pharmacokinetics of the initial molecule.

However, this position did not gain enough attention, as C-7 modifications of FQ have been the most popular in recent years. Still, there are valuable investigations in this area, and their summary is presented in Figure 3 and Table 1. The text in this section provides both original and generalized reaction schemes. Details on synthetic procedures from the literature data are provided in the Appendix A.

To prepare the review, we used the chemical and bibliographic information SciFinder database (https://scifinder-n.cas.org/ (accessed on 13 December 2022)) and PubMed database (https://pubmed.ncbi.nlm.nih.gov/, accessed on 13 December 2022). Marvin was used for drawing, displaying and characterizing chemical structures, substructures and reactions, Marvin 23.1.0, 2023, ChemAxon (https://chemaxon.com/, accessed on 1 February 2023). DataWarrior was used for drawing diagrams, OSIRIS DataWarrior 5.5.0, 2021 (https://openmolecules.org/datawarrior/, accessed on 1 February 2023).

### 1.1. Modification of FQs with Five-Membered Heterocycles

The most pronounced contribution in the investigated area was made by the scientists who worked with the substitution of the C-3 carboxylic acid group with subsequent cyclization, obtaining a wide range of 3-heteroaryl hybrids of initial FQs.

On the one hand, there is evidence of the importance of carboxylic acid group presence for the manifestation of the antimicrobial activity of fluoroquinolones. On the other hand, small heterocyclic moieties (five-membered heterocycles) have already proven themselves to be an effective modification tool for the piperazine ring of such well-known antibiotics as norfloxacin and ciprofloxacin [23,24].

They can be introduced into C-3 via the formation of different functional derivatives of carboxylic acids that are presented in Figure 1. For instance, alkylation offers an opportunity to further obtain hydrazine derivatives [25,26,27,28,29,30,31,32,33,34,35,36,37,38,39,40,41,42,43,44,45,46,47,48,49,50,51,52,53,54,55]. The carboxylic group can be easily turned into an amide group with the subsequent formation of nitriles [56]. Among other common derivatives are acid halides [57,58] and thiosemicarbazides [41,59]. All of them can be prepared easily and used for further cyclization. In addition, several works describe 3-acetyl derivatives obtained directly via the Gould–Jacobs reaction [60,61,62] and then converted into bromoacetyl derivatives; the formation of amines [63] from nitro derivatives is also an interesting case.

As for further cyclization, literature data present investigations that describe synthesized 3-heteroaryl hybrids of fluoroquinolones with oxadiazole [25,26,27,28,29,30,31,32,33,34,35,36,37,38,39,57,64,65,66,67,68,69,70], isoxadiazole [55,68], triazole [25,26,30,40,41,42,43,44,45,46,65,71,72,73,74,75], thiadiazole [25,26,47,59,60,71,75,76,77,78,79,80,81,82], thiazole [26,61,62,83,84,85,86,87], oxazole [25,57] and tetrazole [56,58,63,88,89] rings.

The basic structure of these derivatives is similar, differing only by one heteroatom in the ring. As a result, similar approaches are usually utilized for their synthesis, and the possibilities of converting one heterocycle into another are also used.

One of the leading positions among them is given to the oxadiazole ring that was used for modification of fluoroquinolones of the second (norfloxacin, ciprofloxacin, ofloxacin, pefloxacin) and third (levofloxacin, enrofloxacin) generations [25,26,27,28,29,30,31,32,33,34,35,36,37,38,39,57,64,65,66,67,68,69,70]. The main routes for the introduction of the oxadiazole ring into C-3 of fluoroquinolones are depicted in Figure 2. The first (a) consists of interaction of the corresponding FQs with acyl hydrazides (semicarbazides), and subsequent cyclization in the presence of POCl_3_ or bases [64]. The second variant of this reaction (b) is the interaction of preliminarily obtained hydrazides with aliphatic or aromatic acids in similar conditions [29,35,37,52].

The resulting oxadiazole hybrids are often transformed into other derivatives. When interacting with various amino compounds, the oxadiazole fragment is modified into a 1,2,4-triazole fragment. An example of such a modification is presented in Figure 3 [64]. Target products CT 1.3 are synthesized by refluxing the obtained oxadiazole hybrids with 2,3,4,5-tetrafluorobenzohydrazide in dry pyridine.

A variation of derivatives 1.1 are mercapto/thio substituted compounds 1.2. A popular synthesis method for them consists of the formation of oxadiazolthiones via treatment of synthesized hydrazides with carbon disulfide (Figure 4). The corresponding amino derivatives are obtained by the modification of the carboxyl group of fluoroquinolones by the action of semicarbazide in the presence of a cyclizing agent [25].

Figure 4 also summarizes methods for the introduction of 1,2,4-triazole fragments into FQ molecules. One of them is the conversion of oxadiazoles, as mentioned above. For example, the reaction of compounds 1.3 with hydrazine hydrate yielded FQ containing a 3-thio-4-amino-1,2,4-triazole fragment in the 3 position (CT 1.3B) [25,49,50,53,54].

Similar triazol derivatives were synthesized from corresponding acids or hydrazides via interaction with hydrazine hydrate and carbon disulfide. Thiosemicarbazide reactions are regioselective. Depending on the conditions, the authors synthesized hybrids containing a 3-thio-s-triazole (in the presence of KOH) or 5-aminothiadisole (POCl_3_) fragment (Figure 4) or thiosemicarabzide [25,60] with subsequent cyclization or via cyclocondensation (Figure 4) [45,46,71].

Thiadiazole hybrids can be easily obtained through a one-step procedure, namely, the reaction with thiosemicarbazide and POCl_3_ under heating (Figure 4) [25,26,59,77,79,81].

For example, Lihumis H.S. et al. [25] described a modification of the carboxyl group of ofloxacin using all the hybrids shown in the scheme.

It is worth mentioning that the obtained 3-heteroaryl FQ hybrids (Figure 4) can be further modified via the approaches that are depicted in Figure 5.

Thus, such modifications include: alkylation of the thiol group in oxadiazol [27,30,31,36,48] or s-triazol [30,36,41,60] moieties; condensation with aromatic aldehydes for amino derivatives [45,71,77,82]; and Mannich reaction [45] with formaldehyde and amines.

In the literature data, there are various options for implementing the last reaction for hybrids of FQ with s-triazoles and oxadiazoles. For example, the use of methyl ketones in the Mannich reaction allowed scientists [46] to synthesize new compounds based on ofloxacin (Figure 6).

Schiff bases synthesized from 3-mercaptotriazole-modified levofloxacin were introduced by the authors [71] into the Mannich reaction with formaldehyde and phenols (Figure 7). It has been proven by spectral methods that the reaction conditions proposed by them favor the reaction on the aryl ring with the formation of polymeric products.

The literature data also describe the preparation of the corresponding hydrazide from the product of alkylation of mercaptooxadiazole with chloroacetic ester and its transformation into a series of arylhydrazones (Figure 8). For such transformations, the authors used ofloxacin [27,60] and pefloxacin [36].

Furthermore, to study the effect on antitumor activity, the synthesis of bioisosteric 3-hydrazinotriazole derivatives obtained as a result of sequential modification of the carboxyl group of ofloxacin was carried out [60]. The compounds were also used in interactions with arylaldehydes (Figure 9).

Another direction of ofloxacin modification was the introduction of the 3-mercapto-s-triazole moiety into its molecule (via the reaction with potassium thiocyanate and subsequent cyclization under the action of alkali) and further alkylation with bromoacetophenones. The resulting alkyl derivatives were converted to the corresponding thiosemicarbazones (Figure 10) [41].

The previously synthesized N-butyl analog of FQs with a morpholine fragment at position 7 and a GHQ168 carboxyl group modified into benzylamide was used by scientists [59] for further chemical modification (Figure 11). Based on the starting acid, the corresponding heterocyclic derivatives were synthesized—1,3,4-thiadiazole and 1,2,4-triazole; the modification of the 1,2,4-triazole derivative by maleimides is also presented in this article.

2-Aminothiazole has been identified by scientists as one of the key pharmacophores in the structure of antibacterial medicines [61,62,83]. A plausible method for the construction of a thiazole ring takes place via Hantzsch thiazole synthesis; the thiazole derivative was formed by condensation of initial α-haloketone and thiourea (Figure 12) [61,62,83].

Interestingly, an attempt to introduce a piperazine or a similar fragment into position 7 in order to obtain a norfloxacin analogue resulted in a mixture of amination products. The authors [62] synthesized 2-aminothiazole hybrids with each of the resulting products after their separation.

A few articles describe the incorporation of a tetrazole ring in a fluoroquinolone molecule [79,90]. Namely, the conversion of 3-amino derivatives into tetrazoles is possible (Figure 13) [63].

The authors [63] used Chitosan-supported Cu nanoparticles to modify the carboxyl group of 1-alkyl-6,7-dihalo-4-oxo-quinoline-3-carboxylic acids into a nitro group. After nitration, the piperidine fragment was successfully introduced into the molecule. To restore the nitro group to the amino group, tin chloride was used. The amino derivative reacted with sodium azide and triethylorthoformate, as a result of which tetrazole hybrids of fluoroquinolones were obtained.

Another approach for the formation of a tetrazole ring based on the carboxyl group in the 3d position of tricyclic fluoroquinolones was used in the study [56]. Preliminarily synthesized primary amides were converted into nitriles by the action of POCl_3_. The cyclization of the latter to tetrazoles was carried out by the action of hydroxylamine or sodium azide/zinc chloride (Figure 14).

A similar approach—the action of azide on the corresponding nitrile—was used by the authors of [57] to introduce the tetrazole fragment (Figure 15). In the presented study, the authors synthesized various bioisostere hybrids of FQs for subsequent comparison of their pharmacological activity. Derivatives of 1,3,4-oxadiazole obtained via the interaction of the acid chloride of the initial acid with diphenylacetic acid hydrazide or isomeric derivatives of 1,2,4-oxadiazole were synthesized using amidoxime. Synthesis of 1,3-oxazole hybrids was carried out using two methods, with acid chloride or nitrile as initial compounds.

### 1.2. Modification of the C-3 Carboxylic Acid Group with Fused Heterocycles

Among the published investigations, there are those related to fused heterocyles as C-3 substituents in the initial FQs. They utilize similar synthetic approaches, mainly, the formation of a triazole ring with its subsequent substitution and cyclocondensation or the reacton with o-aminophenol and its analogues.

For example, 12 novel C-3 thiazolotriazole derivatives were synthesized from levofloxacin by the authors of [48,88] according to Figure 16.

Furthermore, the authors of [49,50] reported the synthesis of 1-cyclopropyl-6-fluoro-7-piperazin-1-yl-3-(6-substituted-phenyl-7*H*-[1,2,4]triazolo[3,4-*b*][1,3,4]thiadiazin-3-yl)-quinolin-4(1*H*)-ones. Further ring contraction via a sulfur extrusion reaction led to new tri-acetylated fused heterocycles related to pyrazolo[5,1-c][1,2,4]triazoles (Figure 17). A similar pathway is described in ref. [91] for 4-amino-5-mercapto-3-FQ-3-yl-1,2,4-triazole derivatives of norfloxacin, ciprofloxacin, enrofloxacin, ofloxacin and levofloxacin. Condensation with 4-(chloroacetyl)catechol in sodium hydroxide–ethanol–water solvents at room temperature formed the opening products that were subjected to an acid-catalyzed intramolecular cyclocondensation to yield the corresponding [1,2,4]triazolo[3,4-*b*][1,3,4]thiadiazines. Heating the fused heterocycles in acetic acid readily resulted in a ring contraction of fused six-membered thiadiazine via a sulfur extrusion reaction, leading to the formation of fused five-membered pyrazolo[5,1-*c*][1,2,4]triazoles.

Another C-3 substituted 1-amino-5-mercapto-1,3,4-triazole enrofloxacin derivative was synthesized via cyclocondensation with chloroacetic acid and further modification of the fused core via a Perkin reaction with substituted benzaldehydes with the formation of s-triazolo[2,1-*b*][1,3,4]thiadiazin-3-ones (Figure 18) [50].

Cyclocondensation with isonicotinic acid under microwave irradiation that led to 7-chloro-1-cyclopropyl-6-fluoro-3-[6-(4-pyridinyl)-1,2,4-triazolo[3,4-*b*][1,3,4]thiadiazol-3-yl]-4(1*H*)-quinolinones is described in ref. [47]. Novel fused C-3 thiazolo[3,2-*b*][1,2,4]triazoles (Figure 19), also obtained via a condensation–cyclization reaction, are described by the authors of [92].

A few more articles were devoted to benzimidazoles, benzoxazoles and benzothiazoles as fused derivatives of fluoroquinolones. First and foremost, a thorough molecular modelling study [93] revealed the promising biologically active molecules among these compounds.

A series of novel ciprofloxacin and levofloxacin derivatives were synthesized via condensation following the Mannich reaction (Figure 20) [94].

Quinolone-3-carboxylic acids were condensed with *o*-phenylenediamine, *o*-aminophenol or *o*-aminobenzenethiol in polyphosphoric acid (PPA) at 170–250 °C to obtain a series of fused derivatives that were further nitrated and hydrogenated (Figure 21) [95].

Similar cyclization of a quinolone derivative with *o*-phenylenediamine, 2-aminophenol or 2-aminobenzenethiol in the presence of polyphosphoric acid was reported by the authors of [95,96].

To form hybrids of tetrafluoroquinolones with benzoxazine and quinoxaline, 1-aryl(alkyl)-3-ethoxalyl-5,6,7,8-tetrafluoroquinolin-4-ones were synthesized (Figure 22). Target products were obtained as a result of interaction with *o*-aminophenol or *o*-phenylenediamine, respectively [89]. Via the reaction of compounds 2.7A with morpholine, it was possible to obtain 7-mono and 5,7-disubstituted products, depending on the reaction conditions.

The introduction of the quinazoline fragment was carried out as a result of the interaction of norfloxacin and ciprofloxacin acid chlorides with 5-chloroanthranilic acid (Figure 23). The resulting benzoxazine reacted with 4-(6-methylbenzo[*d*]thiazol-2-yl)aniline with the formation of the corresponding product [58].

At last, esters of 1-(7-Z-1-ethyl-6-fluoro-4-oxo-1,4-dihydroquinoline-3-carbamoyl)-5-X-6,7,8-trifluoro-4-oxo-1,4-dihydroquinoline-3-carboxylic were successfully converted into 1,3,4-oxadiazino[6,5,4-*i*,*j*]quinoline derivatives (Figure 24) [51].

### 1.3. Bis-Fluoroquinolones and FQs with Two Heterocyclic Moieties as a Special Case in the Investigated Area

An appealing idea in the synthesis of bis-fluoroquinolones linked via the heterocyclic moiety in the 3d position appeared to be common among scientific publications related to the investigated topic. Such novel molecules were mostly studied as antitumor agents and revealed promising results.

First of all, synthesis of different symmetrical and non-symmetrical bis-fluoroquinolones via the five-membered heterocycle as a link was described. For instance, the carboxylic acid groups of two molecules of fluoroquinolones were successfully replaced with an 1,3,4-oxadiazole ring, and a series of the C3/C3 bis-fluoroquinolones as their HCl salts were obtained (Figure 25) [52].

Another study described the reaction of norfloxacin with hydrazine hydrate with the formation of a hydrazide intermediate, followed by cyclization with the second norfloxacin molecule in the presence of POCl_3_ to target bis-fluoroquinolone in a moderate yield [97].

Furthermore, a group of Chinese scientists patented several investigations on the synthesis of bis-fluoroquinolone oxadiazole carbamide derivatives of levofloxacin [98], rufloxacin [99], moxifloxacin [100], gatifloxacin [101], lomefloxacin [102,103], fleroxacin [104], norfloxacin [105,106] and ciprofloxacin (Figure 3) [107,108]. Another similar bis-fluoroquinolone series contains thiadiazole urea as a linking moiety for fleroxacin [109,110], ciprofloxacin [111], lomefloxacin [112], moxifloxacin [113], pefloxacin [114], ofloxacin [115], levofloxacin [116] and gatifloxacin [117] (Figure 4).

One more interesting route is the modification of already obtained 3-heteroaryl FQ hybrids with five-membered or fused heterocyclic moieties [38,39,55,70]. This was often carried out via substitution reactions with corresponding halides or amines (Figure 26). Such studies have been carried out for levofloxacin [38], norfloxacin [39] and ciprofloxacin [70].

To improve solubility based on the synthesized bis-oxadiazoles, dimethylpiperazinium iodides were obtained [38,39,70]. The alkylation (Figure 27) is shown in the example of levofloxacin derivatives [38]. 

The s-Triazole ring as an isostere modified by an oxadiazole ring corresponding to the C-3 carboxylic acid group for pefloxacin [43] and ofloxacin [44] resulted in new compounds (Figure 28). Products obtained from pefloxacin were converted into Mannich bases [42].

1,2,4-Triazole/benzothiazole combination 3.7 was designed and synthesized using norfloxacin as a starting substance [82]. In the last stage of the synthesis, equimolar amounts of 2-amino-7-chloro-6-fluorobenzothiazole and preliminary synthesized 3-substituted-4-amino-5-mercapto-1,2,4-triazolo Schiff bases were condensed (Figure 29).

In addition, there are bis-fluoroquinolones created via the fused heterocyclic link, as ref. [53] describes, taking, as an example, the [1,2,4]-triazolo[3,4-*b*][1,3,4]-thiadiazole core. In this study, initial cipro- and levofloxacin molecules were substituted with hydrazine hydrate and then were subjected to acyclo-condensation with carbon disulfide in the presence of an excess alkali–ethanol solution (Figure 30). Obtained oxadiazolethiols were converted into the amino s-triazolethiols using hydrazine hydrate. The following condensation into the target compounds was successful only in the presence of POCl_3_.

Similar C3/C3 bis-fluoroquinolones tethered with a fused s-triazolo[2,1-*b*][1,3,4]thiadiazole were obtained by the authors of [54] and patented by the authors of [118].

## 2. Biological Activity of the 3-Heteroaryl FQ Hybrids

Investigations of the biological activity of the FQ hybrids led to bifurcation and revealed two new potentialities, namely, creation of new potent antimicrobials and novel anticancer medicines. These two groups appear to be the most prominent for now. Still, there are a few more cases of other activity types that are also promising.

### 2.1. Novel FQ Hybrids as Antimicrobials and Antiviral Medicines

Among the core FQ molecules that served for the development and research of new potent antimicrobials, the first place is taken by norfloxacin. The carboxylic group of the initial compound was modified mainly with five-membered heterocyles with further investigation of antibacterial, antifungal and antiviral activities.

For example, a series of 1,3,4-oxadiazoles containing FQ derivatives was synthesized and screened for antibacterial and antimycobacterial properties in ref. [31] (Figure 5). The disk diffusion method revealed potent antibacterial activities against *Staphylococcus aureus*, *Enterococcus faecalis*, *Streptococcus pneumoniae*, *Escherichia coli* and *Klebsiella pneumoniae*. In addition, the obtained norfloxacin derivatives showed antimycobacterial activity against *Mycobacterium smegmatis* H37Rv with minimal inhibitory concentrations (MICs) of 22.35, 16.20 and 20.28 μg/mL. The authors also studied absorption, distribution, metabolism and excretion (ADME) properties and proved the promising pharmacokinetic properties and drug-likeness for the obtained compounds.

It is worth noting that most of the investigations in this area are based on in silico studies of drug-likeness and binding properties. Thus, the authors of [26] described molecular docking investigations (AutoDock Tools-1.5.6) against the receptor GlcN-6P (2VF5) that revealed good binding affinities for synthesized norfloxacin derivatives. In this work, the core molecule was modified with 1,3,4-oxadiazole, thiazolidin-4-one, 1,3,4-oxadiazoline, 1,2,4-triazole and 1,3,4-thiadiazole rings. The antimicrobial assessment via disk diffusion and serial dilution methods showed higher activity than for norfloxacin against *S. aureus*, *Staphylococcus epidermidis*, *Streptococcus pyogenes*, *Micrococcus luteus*, *K. pneumoniae*, *Pseudomonas aeruginosa*, *E. coli* and *Proteus mirabilis*.

1,2,4-Triazole and 1,3,4-oxadiazole norfloxacin hybrids with suitable druglike, antibacterial and antifungal properties were successfully obtained by researches [30]. Especially pronounced was the activity against *S. pneumoniae,* with minimum inhibitory concentrations of 0.89 and 0.96 mg/mL and minimal bactericidal concentrations of 2.95 and 2.80 mg/mL. Combined with simple synthetic approaches, such results are promising for further investigations.

Interestingly, similar oxadiazole norfloxacin and ciprofloxacin derivatives showed good antibacterial activity against both Gram-positive (*S. aureus*) and Gram-negative (*E. coli*) strains, combined with promising antifungal activity against fungi (*Saccharomyces cerevisiae*) in comparison with reference drugs ciprofloxacin and fluconazole in the study [29].

A few interesting works were devoted to an exploration of the antimicrobial potential of aminothiazolyl hybrids of norfloxacin. It was proved that the 2-aminothiazole fragment at the 3-position of the quinolone core plays an important role in exerting antibacterial activity. For instance, in this case, the antibacterial activity investigation revealed higher values in comparison with the reference drugs against methicillin-resistant *Staphylococcus aureus* (MRSA) and *S. aureus* 25923, with MIC values of 0.009 and 0.017 mM [60]. Figure 6 shows the detailed activity and concentrations of the synthesized compounds.

Another series of aminothiazolyl norfloxacin analogs was synthesized by the authors of [71] and was screened for antimicrobial properties. Most of the compounds synthesized were superior to reference drug inhibitory efficiencies against *K. pneumoniae* and *Candida albicans,* with MIC values of 0.005 and 0.010 mM. Furthermore, these compounds revealed better antibacterial activity against *S. aureus* ATCC 29213 and methicillin-resistant strains (Figure 7).

In addition, investigations of the inhibitory activity against the DNA gyrase from *E. coli* showed that aminothiazolyl norfloxacin analogs have good inhibitory potency for DNA gyrase (IC_50_ ¼ 16.7 mM), which was more effective than the reference drug norfloxacin (IC_50_ ¼ 18.6 mM). Altogether, the authors concluded that the replacement of the carboxyl group with the weak basic 2-aminothiazole moiety provides a similar antibacterial mechanism to norfloxacin by targeting DNA gyrase.

They additionally proved their hypothesis via docking studies with the topoisomerase IV-DNA complex and gyrase-DNA complex. Among the found interactions, the sulfur atom in the 2-aminothiazole moiety participated in the non-covalent coordination with ARG-136 residue via hydrogen bond formation that is favorable for stabilizing the compound-enzyme-DNA supramolecular complex, which further accounted for the good inhibitory efficiency against the tested strains.

The investigation in ref [62] also describes a series of novel 2-aminothiazol-4-yl norfloxacin analogs created to combat quinolone resistance. Among them, 3-(2-aminothiazol-4-yl)-7-chloro-6-(pyrrolidin-1-yl)quinolone exhibited potent antibacterial activity, strong inhibitory potency to DNA gyrase and a broad antimicrobial spectrum, including against multidrug-resistant strains. Moreover, this molecule induced bacterial resistance more slowly than initial norfloxacin. The docking evaluation gave good total scores (5.68 and 6.46) for aminothiazolquinolones against topoisomerase IV-DNA and gyrase-DNA complexes.

A tertrazolyl moiety appeared to be one more promising bioisostere that was introduced in the C-3 of norfloxacin and ciprofloxacin [63]. The authors conducted docking studies using a Molegro Virtual Docker (MVD) to prove their idea. The tested compounds showed a very similar binding mode with DNA gyrase compared to the co-crystallized ciprofloxacin. The tetrazole formed three hydrogen bonds with Ser1084 instead of one, which is formed by ciprofloxacin. The bond lengths of the three hydrogen bonds were 1.9, 2.1 and 3.1Å. The nitrogen of the piperazine formed a hydrogen bond with base pair DNA backbone DT4. The MolDock score (kcal mol^−1^) and the Rerank Score were −123.54 and −74.67, respectively. Overall, the tested compound revealed a similar manner to ciprofloxacin, with additional hydrogen bonds related to the 3-tetrazole scaffold, supporting the molecular design.

In addition, this modification led to optimization of the solubility profile of the initial molecules. As for the antibacterial activity, the inhibition zones for *S. aureus* and MRSA were from 12.5 to 25 mM. Several derivatives revealed activity at 12.5 and 25 mM, respectively, against *Salmonella typhi*, while reference drugs were active at 100 mM. Moreover, high activity against *Vibrio cholerae* and *E. coli* was observed.

A few more investigations described modifications of ciprofloxacin. Thus, ref. [32] evidenced the synthesis and evaluation of antibacterial activity of ciprofloxacin C3 hybrids with isatins, phthalimides and oxadiazoles. In vitro antibacterial evaluation was made using disk diffusion and serial dilution methods, and antitubercular activity was measured using the Lowenstein–Jensen (LJ) method. All the obtained compounds were highly effective against *E. coli*, *K. pneumonia* (Gram-negative) and *S. aureus* (Gram-positive) at concentrations of 75 and 100 μg/mL. In addition, they possessed antitubercular activity against normal, multidrug-resistant and extensively drug-resistant strains of Mycobacterium tuberculosis. Therefore, they are more potent antimicrobial agents than ciprofloxacin.

Similar research was conducted by scientists [37] who successfully obtained 1,3,4-oxadiazole hybrids of ciprofloxacin and tested them against the standard group of Gram-positive and Gram-negative microorganisms. Here, again, promising activity was observed for both groups that exceeded the reference drug ciprofloxacin. 

More pronounced activity against Gram-negative strains is described in ref. [65] for new C3 triazole ciprofloxacin derivatives. Antibacterial activity against Gram-positive strains, in this case, remained at the ciprofloxacin level. In addition, molecular docking studies using topoisomerase (3ILW) protein revealed correlation between the antibacterial activity and binding free energy of the molecules. The tested compound showed high affinity with low energy of −6.2 kcal/mol with the employed protein (for ciprofloxacin, it was −6.7 kcal/mol).

Among other core FQs that were utilized, we should mention levofloxacin triazole-3-thiol, oxazole, oxadiazole and thiadiazol derivatives [25]. They were tested against *E. coli* and *S. aureus* and, in the disk diffusion method, exceeded the initial compound.

Furthermore, a series of new ofloxacin analogs was synthesized by modifying it by triazoles [46]. In the first stage of the research, in silico docking studies using Autodock vina 4.0 program were performed. Almost all the compounds used for docking showed a best-fit Root Mean Square Difference (RMSD) value of 0.000 with topoisomerase II (3ILW), and good inhibition, with an affinity range between −7.4 and −6.4 kcal/mol. The obtained data were verified via in vitro antimicrobial screening, where the obtained compounds showed promising activity against *S. aureus*, *S. epidermidis* and *Bacillus subtilis* (MIC 0.125 μg/mL).

Several patents on FQs hybridized via five-membered heterocyles at C3 were obtained by Chinese scientists. They claim antibacterial (against Gram-positive and Gram-negative strains), antifungal and DNA intercalating properties in the obtained thiazole [85] and aminothiazole derivatives [84,86]. In addition, they describe a simple and affordable preparation technique based on available raw materials. Another patent [119,120] describes the preparation of novel hybrids with rhodamine as promising antibacterial agents.

Among fused hybrids, the authors of [94] describe novel benzimidazole–quinolinone derivatives of ciprofloxacin and levofloxacin. All the synthesized compounds revealed promising antifungal activity when compared with the reference drug griseofulvin.

A series of new triazolothiadiazole derivatives of ciprofloxacin was obtained and tested against *S. aureus* and *E. coli* [47]. They showed stronger inhibitory activity against *E. coli* than that of *S. aureus* compared to ciprofloxacin. Therefore, the fused heterocycle-based substituted FQs are valuable for further investigations.

Lastly, a novel class of FQ oxadiazole derivatives that inhibit NS5B polymerase, a key enzyme of the Hepacivirus viral life cycle, is described in ref. [57] that may be the first step toward exploration of this dimension of FQs.

Altogether, the published studies definitely prove the logic and efficiency of these investigations. Even if they are now underestimated, their continuation gives a perfect chance to obtain fruitful results, namely, new potent antimicrobials to combat the problem of resistance to antibiotics.

### 2.2. Novel FQ Hybrids as Promising Antitumor Agents

Surprisingly, literature data revealed that many research groups searched for antitumor agents among new FQ hybrids. This approach is based on the concept of bioisosters that gained popularity in medicinal chemistry in recent years. Namely, the bioisosteric replacement of the carboxylic group with different heterocyclic moieties and synthesis of bis-fluoroquinolones linked via a heterocycle at C3 are two main strategies that are widely presented.

Another valuable point is the variety of cancer types against which the compounds synthesized were tested in the above-mentioned investigations.

Thus, new 1,3,4-thiadiazole derivatives of ciprofloxacin were synthesized and investigated via thorough in silico and in vitro studies (Figure 8). Theoretical and experimental DNA binding research revealed good correlation with human hepatocellular carcinoma (Huh-7) cell line activity. IC_50_ values from Huh-7 cell line studies (25.75 μM) revealed synthesized compounds as potent anticancer agents, promising for further investigations [79].

Other isosteres of the C-3 carboxylic group for the pefloxacin 1,3,4-oxadiazole-thione ring and oxadiazole thione Mannich bases were suggested by the authors of [31]. The obtained compounds were tested in vitro against a liver cancer (Hep-3B) cell line and, according to the results, all the title compounds showed more significant potency than parent compounds. In addition, derivatives of aliphatic amines appeared to be more active than the derivatives of aromatic amines. Furthermore, similar derivatives were reported in patent [36], and it was shown that molecules with an electron-withdrawing group attached on the aryl ring had more potency than compounds with an electron-donating group.

The authors of [39] patented similar oxadiazole norfloxacin derivatives that were screened in vitro against the same liver cancer (Hep-3B) cell line. Evaluation was performed via an MTT assay. The results revealed higher cytotoxicity compared to norfloxacin for fifteen title compounds. Correspondent quaternary ammonium salts exhibited promising anticancer activity with IC_50_ values below 25.0 μmol/L.

Another search for agents against the human hepatoma (Hep-3B) cancer cell line and human pancreatic (Capan-1) cell line was made based on comparative molecular field analysis techniques [40]. Three-dimensional quantitative structure–activity relationship (3D-QSAR) investigations on the antitumor activity of s-triazole sulfide-ketone derivatives of ciprofloxacin and levofloxacin gave the possibility to design four novel molecules with promising anti-tumor activity and to plan further in vitro research.

A series of ciprofloxacin and norfloxacin oxadiazole derivatives was evaluated for their antiproliferative activities against human lung tumor (A549) cell lines. Among them, the most active compound, 1-cyclopropyl-6-fluoro-3-[5-(4-nitrophenyl)-1,3,4-oxadiazol-2-yl]-7-piperazinyl-1,4-dihydro-quinolin-4-one, was found, with a half-maximal inhibitory concentration (IC_50_) of 9.0 μg/mL [66].

Furthermore, the scope of cancer cell lines expands via other organs. For instance, the authors of [46] patented novel ofloxacin 1,3,4-triazole derivatives as antitumor agents for treating bladder, stomach or pancreatic cancer (Figure 9).

Research related to bladder tumors is also patented [73] and describes similar triazole derivatives for which an IC_50_ value of 0.6μM against a bladder tumor was detected.

The series of C-3 s-triazole thioether ketone semicarbazone ciprofloxacin hybrids was synthesized via multi-step synthesis and patented as promising antitumor agents for treating stomach, pancreatic or bladder cancer [72]. Inhibitory activity with an IC_50_ of 6.4, 10.5, 9.7, 3,8, 3.6 and 48.6 μM against bladder cancer (T24), gastric cancer (HGC823, HGC27), pancreatic cancer (Panc-1, Capan-1) and VERO cancer cell lines, respectively, was revealed.

A simple synthetic approach led to the development of novel citrate-triazole-oxadiazole norfloxacin hybrids that exhibited remarkable anticancer activity against cervical cancer (HeLa) cell lines with an IC_50_ value 11.3 ± 0.41, comparable to the standard drug [30]. The compounds also revealed suitable druglike properties and are expected to present a good bioavailability profile.

Bis-lomefloxacin derivatives linked via an oxadiazole-carbazide bridge showed high activity against the human non-small-cell lung cancer (A549) cell line, human pancreatic cancer (Capan-1) cell line and human skin melanoma (A375) cell line, exceeding the parent compound, as well as isomerase inhibitor hydroxycamptothecin (HC) and tyrosine kinase inhibitors Ragofini (RRF) and cabozantinib (CZT) [102,103].

Similar bis-fluoroquinolone oxadiazole carbazide N-methyl-ciprofloxacin derivatives were synthesized, tested against human A549, Capan-I and A375 cell lines; they showed promising activity and were patented [107,108]. In addition, they reduced toxic side effects on normal cells.

As anti-lung cancer and antihepatoma drugs, bis-fluoroquinolone oxadiazole urea N-acetyl norfloxacin derivatives were also patented [105]. In addition, rufloxacin bis-fluoroquinolone oxadiazole urea derivatives revealed inhibiting activities on the human non-small-cell lung cancer (A549) cell line, human liver cancer (SMCC-7722) cell line, human gastric cancer (HGC27) cell line, human pancreatic cancer (Capan-I) cell line, human skin melanoma (A375) cell line and human leukemia (HL60) cell line [99].

Bis-acetylciprofloxacin linked via thiadiazole and urea pharmacophore showed higher antitumor activity and selectivity as well as a reduction in toxic side effects on normal cells [111]. The compounds synthesized were checked on lung cancer (A549), human papillomavirus-related endocervical adenocarcinoma (SMCC-7721), human gastric cancer (HGC27), human pancreatic cancer Capan-1, melanoma (A375), human leukemia (HL60) and myelogenous leukemia (K562G) cell lines, with promising results. N-acetylnorfloxacin thiadiazole norfloxacin derivatives exhibited IC_50_ values in a range of 0.36 μM to 4.66 μM toward cancer cell lines [82].

Many published papers are devoted to the search for potential antileukemic agents among 3-heteroaryl FQ hybrids. For instance, a series of [1,2,4]triazolo[3,4-*b*][1,3,4]thiadiazine and pyrazolo[5,1-*c*][1,2,4]triazole derivatives of norfloxacin, ciprofloxacin and levofloxacin was successfully obtained [91]. Their in vitro antitumor activity was tested against murine leukemia (L1210) and Chinese hamster ovary (CHO) cell lines via the standard MTT assay. The results showed poor inhibitory activity for parent fluoroquinolones (IC_50_ > 150 mmol/L), while isolated fused compounds had a potential activity with an IC_50_ value within 10.0 mmol/L.

Several in silico studies prove this idea. Thus, for the therapy of T-cell lymphoma, a molecular electronegativity distance vector of levofloxacin-thiadiazole histone deacetylase (HDAC) inhibitor was measured [76] and the compounds were described as promising.

Quantitative structure–activity relationship (QSAR) and molecular docking study of levofloxacin and thiadiazole as antitumor agents with HDAC1, HDAC2 and HDAC6 also revealed that the main factors affecting their biological activity are hydrogen bonding and hydrophobic interactions [78]. From the docking results, it can be seen that the active part of the molecule formed a hydrogen bond with the active part of the macromolecule, while the hydrophobic part of the small molecule had a hydrophobic interaction with non-polar amino acid residues in the active part of the macromolecule.

The authors of [121] conducted detailed QSAR analysis of 6-fluoro-3-(4*H*-1,2,4-triazol-3-yl)quinolin-4(1*H*)-ones to explore the features responsible for the selectivity of the antileukemic activity against the human promyelocytic leukemia (HL60) cell line. They created a useful base that further contributed to other research.

For instance, novel oxadiazole [35] and oxadiazolone thione Mannich base derivatives of enrofloxacin [34] were synthesized and studied as promising antitumor agents. Their antiproliferative activity against hepatocellular carcinoma (SMMC-7721), murine leukemia (L1210) and human leukemia (HL60) cell lines was evaluated via an MTT method. The results revealed improved activity compared to enrofloxacin.

Novel C-3 s-triazole-oxadiazole sulfide Mannich base pefloxacin derivatives were also screened against SMMC-7721, L1210 and HL60 cell lines and were evaluated by an MTT assay. The investigation revealed that the sulfides and their corresponding Mannich base compounds are more potent inhibitors than the starting compounds, especially against SMMC-7721 [42,43].

New s-triazole ofloxacine derivatives with functionalized side chains of Schiff bases and Schiff–Mannich bases are described in ref. [45]. Their in vitro antitumor activity against L1210, Chinese hamster ovary (CHO) and HL60 cell lines was evaluated in the MTT assay. Higher inhibitory activity was detected for compounds possessing a free phenol group. Other novel s-triazole-thiosemicarbazone ofloxacine derivatives were also evaluated by the MTT assay and revealed more significant antiproliferative activity than parent ofloxacin. Thiosemicarbazones, especially those containing a nitro group or fluorine atom, showed activity comparable to doxorubicin. Therefore, an azole ring modified with a functional side chain is a favorable bioisosteric replacement of the C-3 carboxylic group for an improvement in antitumor activity [41].

Aminothiadiazole ciprofloxacin derivatives and their Schiff bases were screened against SMMC-7721, HL60 and L1210 cell lines via MTT assay [77]. They showed potential cytotoxicity with IC_50_ values that reached micro-molar concentration and were patented.

Another patent was obtained for the invention of floxacin diazole derivatives that exhibited IC_50_ values of approximately 2, 5 and 2.5 μM against CHO, HL60 and L1210 cell lines, respectively [67].

Several papers describe screening of anticancer activities against CHO, HL60 and L1210 cancer cell lines of fused FQ hybrids. Thus, s-triazolothiadiazole ciprofloxacin derivatives revealed significant antitumor activity against HL60, with IC_50_ values from 50.0 to 8.0 μmol/L [47,49], s-triazolothiadiazine ciprofloxacin derivatives showed more significant inhibitory activity (IC_50_ < 25.0 μmol/L) than parent ciprofloxacin (IC_50_ > 150.0 μmol/L), s-triazolothiadiazinone enrofloxacin derivatives exhibited significant antitumor activity, with a range of micromole concentrations for IC_50_ value [50], and C-3 thiazolo[3,2-*b*][1,2,4]triazole ofloxacin derivatives exhibited more significant antiproliferative activity than parent ofloxacin [92]. Novel C-3 thiazolotriazole levofloxacin derivatives showed more significant activity than levofloxacin [48]. The compounds with fluorophenyl or o-methoxyphenyl displayed comparable activity to doxorubicin. Antitumor agents, prepared by cyclization of norfloxacin with o-phenylenediamine, 2-aminophenol or 2-aminobenzenethiol in the presence of polyphosphoric acid (PPA), showed strong antitumor activity in cows [96]. Therefore, a fused heterocyclic moiety as an isostere of the C-3 carboxylic acid group appears to be an alternative approach for further design of active antitumor FQs.

Moreover, four different bis-fluoroquinolones investigations in this direction were also made. For example, a series of C3/C3 bis-fluoroquinolones tethered with an 1,3,4-oxadiazole ring was screened against L1210, CHO and HL60 cell lines, showed promising inhibitory activity and was patented [52]. 1,3,4-Oxadiazole-linked norfloxacin showed antitumor activity with IC_50_ values of 15.6, 20.5 and 7.6 μM against CHO, HL-60 and L1210 cell lines, respectively [97]. Bis-oxadiazole methylsulfide derivatives derived from ciprofloxacin [70] and levofloxacin [38] were tested against CHO, HL60 and L1210 cancer cells and evaluated by MTT assay. The preliminary results showed that piperazinium compounds possess more potent activity than that of corresponding free bases.

Furthermore, ciprofloxacin cross-linked with a [1,2,4]-triazolo[3,4-*b*][1,3,4]-thiadiazole core as a common bioisostere of two carboxylic acid groups appeared to be highly potent against the HL60 cell line [53]. In vitro antitumor activity of norfloxacin dimers linked with a s-triazolo[2,1-*b*][1,3,4]thiadiazole moiety against L1210 and CHO cell lines was evaluated and appeared to be promising [54]. 1,2,4-Triazolo[3,4-*b*][1,3,4]thiadiazole-linked ciprofloxacin and levofloxacin dimers were prepared in a multi-step synthesis and revealed inhibitory activities with IC_50_ values of 1.1, 0.25 and 0.15 μM against CHO, HL60 and mouse lymphocytic leukemia (L1210) cell lines, respectively [118].

In addition, there are papers devoted to the hybridization of ofloxacin at C-3 with an s-triazole ring [60], oxadiazole-5-sulfanylacetylhydrazone moiety [27], triazole-oxadiazole methylsulfide [44] and oxadiazole ring [37]. All of them state higher antitumor activity than for the parent ofloxacin in the MTT assay. Similar derivatives patented by the authors of [71] gave a CHO IC_50_ value of 1.3 μM. A patent for bis-oxadiazolyl methylsulfides derived from ofloxacin describes in vitro antitumor activity evaluation against three cancer cell lines by the MTT method. The compounds synthesized showed potential anticancer activity (IC_50_ < 25μmol/L). The activity of the quaternary ammonium salts was higher than that of the corresponding free bases [40].

Further, there are many inventions on bis-fluoroquinolones as antitumor agents that were patented. Namely, levofloxacin-containing bis-fluoroquinolone oxadiazole carbamide derivative is described as useful in the treatment of cancer [98]. A novel bis-fluoroquinolone thiadiazole urea-series fleroxacin derivative was developed by the authors of [109] to increase the antitumor activity and selectivity of fluoroquinolones and reduce the toxic side effects on normal cells. Novel thiadiazole urea rufloxacin derivatives exhibited IC_50_ values in a range of 0.46 μM to 2.36 μM [110]. A series of bis-fluoroquinolone thiadiazole urea N-Me lomefloxacin derivatives is described in patent [112], similar bis-fluoroquinolone thiadiazole urea-based N-Me moxifloxacin derivatives designed as promising antitumor agents were patented [113] and the authors continued this project with thiadiazole urea-based pefloxacin [114], ofloxacin [115], levofloxacin [116] and gatifloxacin [117] derivatives. Furthermore, N-Me moxifloxacin-containing bis-fluoroquinolone oxadiazole urea derivatives for cancer treatment are proposed in the patent [100], N-Me gatifloxacin-containing bis-fluoroquinolone oxadiazole urea derivative in [101], oxadiazole urea fleroxacin derivatives with IC_50_ values from 0.14 μM to 1.36 μM in [104] and oxadiazole urea pefloxacin bis-derivatives in [106].

At last, there are two unusual papers that are unique according to their research strategy. Thus, a novel series of topoisomerase I (Top I) inhibitors was designed via condensation of FQs with *o*-phenylenediamine, *o*-aminophenol or *o*-aminobenzenethiol in polyphosphoric acid (PPA). The most potent compound 1-ethyl-3-(6-nitrobenzoxazol-2-yl)-6,8-difluoro-7-(3-methylpiperazin-1-yl)-4(1*H*)-quinolone revealed a significant inhibitory effect on Top I, leading to Top I-mediated cleavage and influencing Top I expression at the cellular level. Moreover, it induced cell death via apoptosis and accelerated DNA strand breaks without significant alteration in cell cycle populations. The in vivo evaluation on the growth of HT-29 tumor xenografts in nude mice showed its therapeutic potential for further development [95].

The authors of [93] searched for P-glycoprotein ABCB1 inhibitors among fused FQ derivatives. The ABCB1 is involved in multidrug resistance of tumor cells by preventing intracellular accumulation of cytotoxic drugs. In addition, its overexpression limits drug oral bioavailability. To find new potent ABCB1 inhibitors, a 3D pharmacophore model was created based on known inhibitors. The inhibitory activities of the best hits were evaluated by several biological assays, such as rhodamine 123 accumulation assay, chemosensitization assay and multidrug resistance 1-Madin-Darby canine kidney cell/Madin-Darby canine kidney cell permeability assay. The most promising compounds were identified and taken for further development.

Altogether, we can conclude that the scope of antitumor activity of FQ derivatives is quite broad and promising.

### 2.3. Other Types of Biological Activity

Apart from antimicrobial and anticancer potency of FQs, we should emphasize the possibility to broaden the horizons of their utilization as biologically active molecules. This idea was proved by several noteworthy investigations presented below.

First of all, in line with the problem of combating infectious diseases, the authors of [59] searched for potent molecules to cure protozoal infections. It was already known that N-benzylamide derivative of norfloxacin is promising so they synthesized thiosemicarbazide of 1-butyl-6-fluoro-7-morpholino-4-oxo-1,4-dihydroquinoline-3-carboxylic acid and its heterocyclic derivatives—1,3,4-thiadiazole and 1,2,4-triazole. Furthermore, modification of the 1,2,4-triazole ring with maleimides was also described. For the obtained compounds, antitrypanosomal activity was also typical but lower than for N-benzylamide derivative. Still, the C3 substitution plays a key role in this type of activity.

Triazole derivatives of norfloxacin and their Schiff bases were successfully obtained and described in the paper [82]. The Schiff bases (Figure 10) that were screened for analgesic and anti-inflammatory activity on the carrageenan-induced rat paw edema model revealed encouraging results and exceeded the reference drug ibuprofen. At the same time, unsubstituted triazoles revealed antibacterial and antifungal activity. Therefore, further investigations in this area may result in compounds with a versatile pharmacological profile.

The synthesized levofloxacin triazole-3-thiol, oxazole, oxadiazole and thiadiazol derivatives revealed antioxidant activity equivalent to ascorbic acid (IC_50_ = 31.95 g/mL) in the investigation [25].

A series of novel benzimidazole-quinolinone derivatives of ciprofloxacin and levofloxacin was screened for in vitro antidiabetic activity by α-glucosidase inhibitory action and appeared to be promising at a 200 μg/mL concentration compared to acarbose [94].

Novel 5-amino-1,3,4-thiazidazole hybrids of norfloxacin and levofloxacin were synthesized, characterized and assessed for their acetyl cholinesterase enzyme (AChE) inhibitory activity [81]. The obtained derivatives showed promising results, especially levofloxacin derivative (IC_50_ 18.1 ± 0.9 nM), which substantially exceeded the reference drug neostigmine (IC_50_ 2186.5 ± 98.0 nM). In addition, the authors evaluated the ADMET parameters, and values of the hybrids showed appropriate correlation with the binding energy values (Kcal/mol). Combined with high drug-likeness scores and the results of molecular docking studies, this research can be a promising background for finding a treatment for Alzheimer’s disease.

Altogether, we can assume that the scope of the probable biological activity of FQ hybrids is wide. That makes this area of investigation even more versatile and attractive for medicinal chemists.

Table 1 summarizes all the data on the chemotypes that were utilized for the structural modification of FQs as well as the activities studied for the synthesized compounds.
antibiotics-12-00625-t001_Table 1Table 1The general scope of the chemical synthesis and chemotherapeutic potential of 3-heteroaryl fluoroquinolone hybrids.ChemotypeSchemeFQ *Activities Studied **
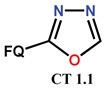
Figure 2 and Figure 4(-)OF [25] aminoAM [25], AO [25]CF [29,64,66,70]AM [29], AF [29], AT/MTT [66]NF [26,28,29,64,66]AM [26,29], AF [26,29], AT/MTT [26,28,66] EF [35]AT/MTT [35]OF [37,69]AT/MTT [37]FQ analogues [29]AM [31], AF [29]Figure 15FQ analogues [57]AV [57]
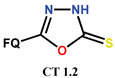
Figure 4, Figure 5, Figure 6 and Figure 7PF [31,36]AM [31], ATB [31], AT [36]NF citrate [30]AC (HeLa) [30], AM [30], AF [26,29,30]CF [32,37,49]AM [32,37], ATB [32]OF [27,46]AT/MTT [27], AM [46]EF [34,50]AT/MTT [34](-)OF [38,67]AT/MTT [38,67]NF [39]AT/MTT [39]
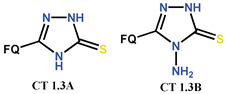
Figure 3, Figure 4, Figure 5 and Figure 6 and Figure 8, Figure 9, Figure 10 and Figure 11(-)OF [25,40,48,67,74,88]AM [25], AO [25], AT/MTT [67,74]NF [26,27,82]NF citrate [30]AM [26,27,30,82], AF [26,30,82], AT/MTT [26,30], AI [82], ATB [27]CF [27,49,65]CF,N-Me [72]AM [27,65], AT [72], ATB [27]OF [41,44,45,46,60,73,92]AT/MTT [41,45,58,60], AM [46]PF [42,43]-EF [50]-GHQ168 [59]ATr [59]
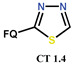
Figure 4 and Figure 5(-)OF [25,67,81](-)OF, desmethyl [81]AM [25], AO [25,81], AT/MTT [67], AChE inhibitors [81]NF [26]NF, N-acetyl [80]AM [26], AF [26], AT/MTT [26,80]CF [77,79,81]AT [77,79]; AO [81], AChE inhibitors [81]GHQ168 [59]ATr [59]
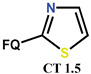
Figure 12NF [61,62,83,84,85,86,87]AM [61,62,83,84,85,86], AT [83], AF [84,86]CF [84,85,86]AM [84,85,86], AF [84,85]
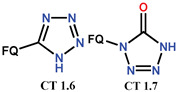
Figure 13, Figure 14 and Figure 15FQ analogues [56,57]AV [57], AM [56], GSK-3b inhibitor [56]NF [63]AM [63]CF [63]AM [63](-)OF [90]-
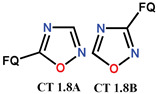
Figure 15FQ analogues [55,57]AV [55,57]
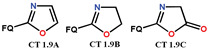
Figure 4(-)OF [25]AM [25], AO [25]Figure 15FQ analogues [57]AV [57]
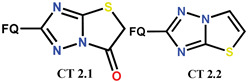
Figure 16(-)OF [48,88]AT [48,88]OF [92]AT [92]
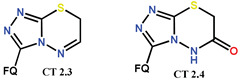
Figure 17 and Figure 18CF [49,91]AT/MTT [49,91]OF [91]AT/MTT [91]EF [50,91]AT/MTT [50,91]
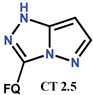
Figure 17CF [49,91]AT/MTT [49,91]OF [91]AT/MTT [91]EF [91]AT/MTT [91]
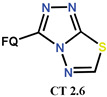
Figure 19CF [47]AM [70], AT/MTT [47]
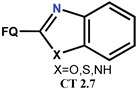
Figure 20 and Figure 21CF [94]ABCB1 inhibitors [93], AD [94], AF [94]OF [94]ABCB1 inhibitors [93], AD [94], AF [94]NF [95,96]ABCB1 inhibitors [96], AT [95,96]
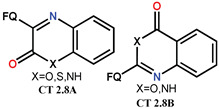
Figure 22 and Figure 23FQ analogues [89]-NF [58]AM [58], ATB [58], AF [58]CF [58]AM [58], ATB [58], AF [58]
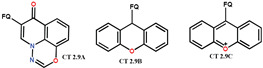
Figure 24NF analogues [51]-FQ analogues [119,120]AM [119,120]
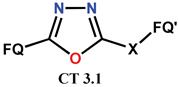
Figure 25NF [52,97,105]AT/MTT [52,97,105], AM [97]EF [52,97,105]AT/MTT [52,97,105], AM [97]CF [52,97,105]CF, N-Me [107]CF, N-acetyl [108]AT/MTT [52,97,105,107,108], AM [97]PF [52,97,105,106]AT/MTT [52,97,105,106], AM [97]OF [52]AT/MTT [52](-)OF [52,98]AT/MTT [52,98]RF [99]AT/MTT [99]MF, N-Me [100]AT/MTT [100]GF, N-Me [101]AT/MTT [101]LF [102,103]AT/MTT [102,103]FL [104]AT [104]
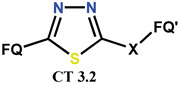

FL [109]AT [109]RF [110]AT [110]CF, N-acetyl [111]AT/MTT [111]LF, N-Me [112]AT/MTT [112]MF, N-Me [113]AT/MTT [113]PF [114]AT/MTT [114]OF [115]AT/MTT [115](-)OF [116]AT/MTT [116]GF, N-Me [117]AT/MTT [117]
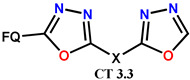
Figure 26 and Figure 27(-)OF [38,67]AT/MTT [38,67]NF [39]AT/MTT [39]OF [69,75]AT/MTT [69,75]CF [70]AT/MTT [70]
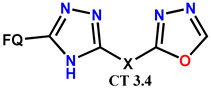
Figure 28(-)OF [67]AT/MTT [67]PF [42,43]AT/MTT (SMMC-7721, L1210 and HL60) [42,43]OF [44,75]AT/MTT [44,75]
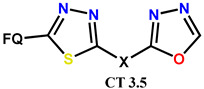

(-)OF [67]AT/MTT [67]OF [75]AT/MTT [67]
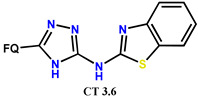
Figure 29NF [82]AI [82], AF [82], AM [82]
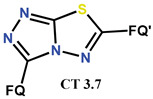
Figure 30CF [53,54,118]CF, N-Me [53]AT/MTT [53,54,118]EF [53,54,118]AT/MTT [53,54,118]NF [53,54,118]AT/MTT [53,54,118]OF [53,118]AT/MTT [53,118](-)OF [53,118]AT/MTT [53,118]PF [54,118]AT/MTT [54,118]* CF—ciprofloxacin, EF—enrofloxacin, FL—fleroxacin, GF—gatifloxacin, (-)OF—levofloxacin, LF—lomefloxacin, MF—moxifloxacin, NF—Norfloxacin, OF—Ofloxacin, PF—Pefloxacin, RF—Rufloxacin, ** AC—anticancer, AI—anti-inflammatory, AM—antimicrobial, AnA—analgesic, AO—antioxidant, AT—antitumor, ATb—antitubercular, ATr—antitrypanosomal, AV—antiviral.

## 3. Conclusions

Fluoroquinolones, nowadays, belong to the medicinal sector. This is true both for clinical practice and for scientific investigations. While all four generations are widely used for treatment of different infections, the issue of resistance of microorganisms becomes demanding as well as their pharmacokinetics improving.

Therefore, medicinal chemists all over the world continue investigations on FQs structural modifications. As the current research revealed, there are promising studies in the area of 3-heteroaryl hybrids. The latter can be synthesized via different convinient methods with the formation of new derivatives with five-membered and fused heterocycles or creation of bis-fluoroquinolones with variable linking moieties. These novel compounds revealed not only good antimicrobial properties compared to the parent molecules but were also widely investigated as anticancer agents with promising activity.

Among the described novel molecules, there are potent antibacterial agents that possess activity against clinical and resistant strains. A big part of these investigations is supported by molecular docking studies that reveal all the nuances of interactions with proteins and binding modes. As for the anticancer activity, we observe a wide variety of cancer cell lines against which the compounds were tested and showed promising results. Moreover, there are studies devoted to potent 3-heteroaryl FQ hybrids as antiprotozoal, analgesic and anti-inflammatory, antioxidant, antidiabetic agents and even acetyl cholinesterase enzyme inhibitors. Furthermore, the big scope of patents obtained serves as proof of interest in this area and prospects for continuation of these studies.

Altogether, we can conclude that this scientific direction is gaining momentum and further investigations on chemical and pharmacological potential of 3-heteroaryl hybrids of fluoroquinolones will be of a great value.

## Data Availability

All data generated or analyzed during this study are included in this article.

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
