# Peer review of "A Comprehensive Review on Chemical Synthesis and Chemotherapeutic Potential of 3-Heteroaryl Fluoroquinolone Hybrids"

_antibiotics, 2023, doi:10.3390/antibiotics12030625_

Round 1
Reviewer 1 Report
In the present manuscript, entitled "A Comprehensive Review on Chemical and Pharmacological Potential of 3-Heteroaryl Fluoroquinolone Hybrids," the authors have a helpful intention: to display the fluoroquinolones' newly derived compounds obtained through hybridization. They presented the aim of their work as follows (lines 59-63).
"In the current review, the results of investigations on the structural modification of FQs by introducing heterocyclic moieties into position 3 of the core system are analyzed and summarized, as well as approaches to the formation of heterocycles from the fluoroquinolone carboxyl group are showed, and the results of a study of the biological activity of the obtained hybrid-pharmacophore systems are presented."
The MS is structured as follows:
Abstract (lines11-22)
1. Introduction (lines 26-63)
2. Synthetic approaches toward 3-heteroaryl fluoroquinolone hybrids
2.1. Modification of FQs with five-membered heterocycles (lines 80-242)
2.2. Modification of the C-3 carboxylic acid group with fused heterocycles (lines 243-325)
2.3. Bis-fluoroquinolones and FQs with two heterocyclic moieties as a special case in the investigated area (lines 326-392)
3. Biological activity of the 3-heteroaryl FQ hybrids
3.1. Novel FQ hybrids as antimicrobials and antiviral medicine (lines 400-542)
3.2. Novel FQ hybrids as promising antitumor agents (lines 544-757)
3.3. Other types of biological activity (lines 759-797)
4. Conclusions (lines 799-812)
The whole manuscript has 30 Schemes and 11 Figures. The Supplementary material presents "Synthesis methods of 3-Heteroaryl Fluoroquinolone Hybrid Сhemotypes" in the other 25 pages.
The reviewer appreciates the hard work of the authors to write such a comprehensive MS and shows some comments below:
Major comments:
The authors are encouraged to synthesize in a few tables the comprehensive information of their manuscript.
1. They could restructure their material, presenting in Introduction the most known FQs, as the base of nowel syntheses and a brief paragraph regarding pharmacological aspects.
They are encouraged to show the databases used for this comprehensive review, the keywords, the inclusion/exclusion criteria, and the methods used for data analysis.
2. In a first table, for example, they could categorize all presented synthesis processes, finding common aspects as follows: the common FQ, the reaction type, the main groups inserted, the structural type of resulted compounds, etc.
3. Another table could include the antimicrobial activity, considering the synthesized compounds, susceptible microorganisms, the method used, the inhibitory concentrations, the inhibition zone diameter, etc...
The antitumor activity should be presented similarly, indicating if the studies are in vitro or in vivo, the synthesized compounds, tumor cell line, cancer type, mechanisms of action, and active concentrations with suitable references.
They should proceed in the same manner showing other bioactivities.
The most important data registered in the tables could be discussed in the MS text, grouped into different classes of new derivatives. The authors should select from all Schemes the most relevant ones.
4. The authors are encouraged to discuss all displayed information in the Discussion section, comparing the structural aspects and pharmacological effects.
5. After all, the conclusions could be concisely presented, eventually showing the most active classes of new hybrids, FQ-derived.
6. Therefore, the abstract could also be restructured with more concrete data.
7. The authors should verify the right to put identical figures from previously published articles in their MS; for example, Figure 7.
8. The authors should verify the concordance between the reference mentioned in the MS and the real ones; for example, Figure 6 - Antimicrobial activity of amino thiazolyl hybrids of norfloxacin - with reference [68], which has no data regarding the antibacterial activity (line 462).
Minor comments:
1. The authors are invited to check and edit the references according to MDPI instructions for authors.
2. The microorganisms' names should be written in italics.
3. The entire MS should be rigorously revised to correct all misprints; for example, lines 441, 462, 463, 466, 470, 480, 481, etc.
4. Figure 8 is unclear.
5. In all figures, the authors should explain their uncommon notations and abbreviations (Figures 5-8).
6. IC50 is better than IC50 (lines 674, 677, etc)
The authors should select the most suitable and directly related to their MS from all references.
Author Response
Dear Reviewer!
We appreciate your contribution and your useful notes and provide the following suggestions:
- In the Introduction section, we added the most known FQs, as you suggested and a few paragraphs about their types and pharmacological aspects.
- The databases used for this comprehensive review, the keywords, the inclusion/exclusion criteria, and the methods used for data analysis.
- The summarizing table has been added that refers to all the mentioned chemotypes and FQs that were used for their introduction as well as references to the schemes and types of biological activity investigations that were made.
- The conclusions were edited.
- The references were checked and edited.
- All the minor points are edited such as the microorganisms' names are written in italics, all misprints are fixed, IC50 is written instead IC50.
Thank you again for your kind remarks. We believe that these changes made the manuscript better.
Best regards,
Authors.
Reviewer 2 Report
In general, the paper is nicely organized, it is easy to follow the idea of the manuscript and the presented matter is interesting.
I have several comments/suggestion/questions for the authors.
1. The spacing of the manuscript should be uniform for entire manuscript- I believe that the first part (Introduction) is different from other parts.
2. All the names of bacteria should be defined at first appearance, and further on used in shortened form. eg. Staphylococcus aureus and further on, S. aureus-- as this is not the case in presented manuscript.
3. Also, all abbreviations should be defined in first appearance, including those well-known such as MIC, MBC, MRSA, VERO etc.
4. Please clarify if all the figures are cited, or are they created by authors of this manuscript.
5. Also, there are many inventions on bis-fluoroquinolones as antitumor agents that were patented.--- Please add some details on clinical use of these derivates if there are available data. If there are other data on any other application (antibacterial), please add these, as application in patients is the ultimate goal of all researches.
6. After reviewing this paper, I highly suggest and request the change of title into A Comprehensive Review on Chemical Synthesis and Chemotherapeutic Potential of 3-Heteroaryl Fluoroquinolone Hybrids, as I believe it better describes the topic which was presented in paper.
7. Not mandatory, but I would like to see the structures of original, non modified FQ - potentially with the comments on their drug-likeness
8. It would be interesting to present a separate section of safety/toxicity of these compounds.
Author Response
Dear Reviewer!
We appreciate your contribution and your useful notes and provide the following suggestions:
- The spacing is made uniform.
- The names of bacteria are given as was recommended.
- All abbreviations are defined in first appearance.
- All the schemes and figures were created by the authors of the manuscript.
- Unfortunately, the papers that we analyzed did not present the details on clinical use of the obtained derivates.
- We changed the title according to the advice given.
- At the beginning we added the structures of original, non-modified FQs
- As far as we conserved there were no sufficient data on safety and toxicity of the synthesized compounds.
Thank you again for your kind remarks. We believe that these changes made the manuscript better.
Best regards,
Authors.
Reviewer 3 Report
The review paper on the synthesis and pharmacological properties of 3-heterocyclic hybrid combinations of fluoroquinolones received for evaluation is extremely interesting and needed by researchers dealing with this subject. It perfectly inspires to further similar research involving also other groups of medicinal compounds.
Fluoroquinolone antibiotics have been known for several decades and are still in the phase of intensive development and the search for numerous new derivatives belonging to their next generations. These derivatives can be of very different kinds and they can be substituted in different positions of their base system. The C-3 carboxyl group has the greatest potential for the formation of differently active derivatives. Compounds of this structure was the main subject in the reviewed article.
And although there are many works on various combinations of fluoroquinolones, this review article is completely original and innovative. It gives a summary of research on currently very fashionable hybrid compounds with a specific structure, which is precisely indicated by the authors in this work. Numerous original experimental papers, as well as respective patent literature, were included in the developed article.
The evaluated work meets all the requirements for good review papers considered in all respects. It has a precisely defined thematic scope and in this respect it is prepared in a fully exhaustive way. The collected data is presented in a very good linguistic and graphical way. It is written in simple and easily understood language, and also in clear and concise manner.
With this great amount of praise, however, some shortcomings and mistakes that need to be corrected should also be pointed out. They are, in the order in which they appear in the text:
1. In line 41 one word "and" is redundant.
2. In line 44/45 it says "a carboxyl group at 3 position" with reference to Fig. 1 A, but in this figure in position 3 is R4. In my opinion, there should be a COOH group, and consequently also the beginning of the sentence in lines 59-61 should be adapted to it.
3. Starting from Fig. 2 and throughout the next parts of the work, I suggest emphasizing the key fluoroquinol moiety marked as FQ. I propose to put these letters on the background of a colourful small circle, which in my opinion will significantly increase the readability of all figures and schemes.
4. The citation of references is usually done according to the scheme of listing them in the order in which they are referenced in the article text. An alternative is alphabetical order. However, none of these methods were used here. This is especially conspicuous in lines 92-93, and is consistently the same throughout the work. I am aware that the change in this respect will not be an easy task, but authors should give the reader the opportunity to at least understand their ideas regarding the principle of quoting individual publications.
5. Based on Fig. 1, it can be assumed that the FQ symbol will be something constant for the whole work. Unfortunately, this is not the case and new designations for FQ appear in Scheme 3. The same is true for many other schemes. In my opinion, FQ should be left constant in accordance with Fig. 1, and if it is necessary, only describe the changes of individual substituents within this FQ should be presented.
6. In all diagrams, sulphur and fluorine atoms are difficult to read due to their colour. I suggest staying with the colour itself to increase its intensity.
7. Scheme 8 can be greatly simplified because the CT 1.8B structure appears in it twice next to each other, so they can be reduced to only one.
8. A very long space is redundant on line 264.
9. In Scheme 21, in the aniline derivative formula, the substituent should be “HX-“ but not "XH-".
10. In the same Scheme, the meaning of the symbols X and Y requires explanation.
11. A new unspecified symbol FQ' appears in Scheme 25. It should be added at the bottom of this Scheme that the indicated structures can be FQ and FQ'.
12. The new symbol FQ2 appears in Fig. 3. It requires an explanation. Drug names are not enough, there should be presented formula for FQ2.
13. In line 340 should be word "describes".
14. In section 3.1. concerning microbiology research, all names of microorganisms should be written in italics.
15. In line 414 there is a double word "are". In my opinion it should be "area are".
16. In the Fig. 5. does not explain the meaning of R1. I believe that here you can give a specific formula of the whole molecule, without using R1, because here are presented properties of specific drugs. At the same time, the meaning of the small structure at the bottom of this Figure is unclear.
17. Line 470 does not contain a drawing number.
18. An invalid character of "Angstrem" was used on line 480.
19. Attached additional materials in this form are not necessary, because all the information contained therein is used in the work itself. But they cover the topic in a slightly different way, so they can remain as a supplement, as it is now.
Contrary to appearances, this large number of suggested corrections does not negatively affect on the very good quality of the reviewed work. For the most part, these are suggestions for editorial corrections, but not substantive ones. Therefore, after taking them into account or responding to them, I suggest accepting the work for publication, without repeating its reviewing process.
Author Response
Dear Reviewer!
We appreciate your contribution and your useful notes and provide the following suggestions:
- We have edited all the minor mistakes and misprints.
- The references are now listed according to the mentioning in the text.
- The summarizing table has been added that refers to all the mentioned chemotypes and FQs that were used for their introduction as well as references to the schemes. We were aimed not to overload schemes with structural formulas and hope that this way it will be convenient to navigate in the manuscript.
- In section 3.1. concerning microbiology research, all names of microorganisms are written in italics.
Thank you again for your kind remarks. We believe that these changes made the manuscript better.
Best regards,
Authors.

Reviewer 4 Report
Dear Authors,
I have reviewed your manuscript, and I am expressing my positive feedback. I believe your review is an important contribution to the field and it will be helpful for the readers of the Antibiotics journal. I gave my best to identify the parts of your manuscript that could be improved, and I managed to generate only a few technical comments:
· Please mention and cite which computational tool(s) was/were used for generating 2d structures of mentioned molecules.
· In Figure 8 the text and numbers are hardly visible because of the white background. It might be useful to add the black background layer to better visualize the names and numeric values of parts of the protein.
· The quality of the text in Figure 9 could be improved. The resolution seems to be too low, while there is also missing a space between the label of the x-axis and the corresponding unit.
· This is a great review paper, but the conclusion is too weak. It should better emphasize the importance of this paper and the considered class of compounds.
Once you address all of the above-mentioned comments, I will gladly review your manuscript again.
Best regards
Author Response
Dear Reviewer!
We appreciate your contribution and your useful notes and provide the following suggestions:
- We mentioned and cited computational tool that was used for generating 2d structures of mentioned molecules before the Figure 3.
- The Figure 8 was deleted from the manuscript due to the copyright issues.
- The quality of the text in Figure 9 (now Figure 8) was improved.
- The conclusion was edited.
Thank you again for your kind remarks. We believe that these changes made the manuscript better.
Best regards,
Authors.
Round 2
Reviewer 1 Report
The reviewer highly appreciates the authors' efforts in their manuscript revision. This current form is substantially improved and suitable for publication in Antibiotics Journal.